# Broiler Mobility Assessment via a Semi-Supervised Deep Learning Model and Neo-Deep Sort Algorithm

**DOI:** 10.3390/ani13172719

**Published:** 2023-08-26

**Authors:** Mustafa Jaihuni, Hao Gan, Tom Tabler, Maria Prado, Hairong Qi, Yang Zhao

**Affiliations:** 1Department of Animal Science, University of Tennessee, Knoxville, TN 37996, USA; mjaihuni@vols.utk.edu (M.J.); gtabler@utk.edu (T.T.); mprado@utk.edu (M.P.); 2Department of Biosystems Engineering, University of Tennessee, Knoxville, TN 37996, USA; hgan1@utk.edu; 3Department of Electrical Engineering and Computer Science, University of Tennessee, Knoxville, TN 37996, USA; hqi@utk.edu

**Keywords:** broiler, welfare, mobility, YOLOv5, semi-supervised learning, neo-deep sort

## Abstract

**Simple Summary:**

Tracking the movements of chickens is important for understanding their well-being. Traditional methods for measuring chicken mobility are time-consuming and cannot provide real-time information. In this study, we, the researchers, used a combination of artificial intelligence methods and computer algorithms to track individual chickens. Using these methods, it was possible to detect and track individual chickens in two pens. We analyzed the data to see how far and how fast each chicken moved every hour and every day. Compared to manual measurements, the combined model provided more accurate measurements of the mobility of each chicken and the entire group, even when some chickens were hidden, or the detection was not perfect. This study may serve to effectively track indicators critical for broiler production performance and welfare.

**Abstract:**

Mobility is a vital welfare indicator that may influence broilers’ daily activities. Classical broiler mobility assessment methods are laborious and cannot provide timely insights into their conditions. Here, we proposed a semi-supervised Deep Learning (DL) model, YOLOv5 (You Only Look Once version 5), combined with a deep sort algorithm conjoined with our newly proposed algorithm, neo-deep sort, for individual broiler mobility tracking. Initially, 1650 labeled images from five days were employed to train the YOLOv5 model. Through semi-supervised learning (SSL), this narrowly trained model was then used for pseudo-labeling 2160 images, of which 2153 were successfully labeled. Thereafter, the YOLOv5 model was fine-tuned on the newly labeled images. Lastly, the trained YOLOv5 and the neo-deep sort algorithm were applied to detect and track 28 broilers in two pens and categorize them in terms of hourly and daily travel distances and speeds. SSL helped in increasing the YOLOv5 model’s mean average precision (mAP) in detecting birds from 81% to 98%. Compared with the manually measured covered distances of broilers, the combined model provided individual broilers’ hourly moved distances with a validation accuracy of about 80%. Eventually, individual and flock-level mobilities were quantified while overcoming the occlusion, false, and miss-detection issues.

## 1. Introduction

The poultry industry growth continues skyrocketing globally, concurrent with the world population surge and the resultant overwhelming demands for ample, nutritious, and affordable protein sources. Meanwhile, this industry is struggling to create an ecosystem that would ensure sustainability and growth by reducing poultry production losses due to health and welfare issues, creating feasible livestock environments through precision technologies, and alleviating labor shortages in giant poultry markets, such as the USA [1].

Welfare plays a commendable role in rendering qualitatively healthy and quantitatively superior poultry productions [2]. Fundamentally, it is a multifaceted phenomenon that demonstrates chickens’ physical conditions, living habitat feasibility, and mental situations. Birds require spacious environments to move with ease, access free feed and water sources, and socialize, while inhibiting infections, lameness, and stress throughout the rearing period. In this context, mobility tremendously affects individual birds’ welfare levels [3]. Impaired or zero locomotion in birds may indicate issues such as insufficient nutrient consumption needed for growth, possible existing pain or stress, lameness, housing constraints, and even mortality. Hence, impaired locomotion has been highly identified in about 15–28% of birds in poultry farms [4]. Hence, mobility or locomotion studies play an important role in shedding light on the conditions that are effective factors for determining individual and flock-level welfare in birds.

The traditional Gait Scoring (GS) and Kinematics (GK) methods have been widely applied to assess locomotion status in individual chickens [5]. While the former method needs to be carried out by an expert tasked to observe and determine gait level one chicken at a time, the latter method utilizes statistical and algorithmic approaches to extract locomotion features, like walking ability, sitting, and standing postures, and correlates them with the predefined GS levels. As in the works of Aydin, Periera et al., and Doornweerd et al. [6,7,8], such features of individual birds are studied and correlated with their predetermined GS levels. Their results are influential in determining major lameness and locomotion problems in individual broilers. Due to welfare management urgency, it is more effective to provide timely, conclusive, subjective, and economical insights into the overall activities of individuals and flocks day in and day out.

On the other hand, several alternative artificial intelligence (AI) and/or Deep Learning (DL) methods have been proposed to tackle the locomotion problem and provide further information on other activities of chickens. The authors Nasiri et al., Fang et al., and de Alencar Nääs et al. [9,10,11] have developed very effective pose estimation and speed-based lameness and behavior detection methodologies using DL models. They have achieved high accuracies in correlating lameness GS levels and behavior classifications with broilers’ skeletal positions and visual analysis, but we still need highly generalizable and practical methods in labs and commercial poultry farms; for instance, cameras used here capture lateral images of flocks, which greatly reduces the analyzability of individual chickens. Lin et al. and Fang et al. [12,13] have applied a Convolutional Neural Network (CNN)-based DL model to detect individual chickens in a shallow setup; consequently, their movements were calculated manually matching consecutive frames. In the work of Neethirajan [14], a CNN-based YOLOv5 model along with a deep sort algorithm were applied to track individual birds’ trajectories and provide periodical movements. Although the study provided flock-level movement orientations, a detailed approach is more helpful in solving the occlusion problem and the consequent individual bird movement and identification issues. These studies are hardly generalizable to broiler farms where constant occlusions happen. Continuously tracking individual bird trajectories and flock-level commotions are thoroughly complicated. The utilization of large hierarchical datasets that would encompass different broiler growth phases and render a less biased DL model would also be crucial in such applications. Additionally, DL has been mostly applied in image classification tasks, as depicted in the previous works. The efficacy of DL models in image segmentation and object detection tasks is yet to be discovered.

Generally, the inadequacies in the quantity of data for developing DL models and the persistence of the occlusion-related identification problems greatly inhibit clear calculations of the individual birds’ trajectories. On the other hand, to achieve economically beneficial and effective applications, one needs to consider the processing time, human resource requirement, and the inherent objectivity of the results. Henceforth, we aimed to tackle the mobility estimation of broilers while handling the occlusion instances, developing a more generalizable and robust model, i.e., the YOLOv5 deep sort model, and a new algorithm for large poultry farms while utilizing large datasets for DL model development. YOLOv5 is a vastly applied object detection DL model for computer vision. It is one of the most advanced versions of the YOLO family, surpassing older YOLO versions in terms of object detection precision. The YOLO models are primarily designed to perform object localization and classification in a single-stage regression process, thus outperforming counterpart CNN-based models, such as faster region-based or mask CNN, in terms of inference speed and memory efficiency, as well as provide comparable precision levels. Therefore, it is deemed one of the most effective models for real-time applications [15]. Meanwhile, the deep sort algorithm has been widely applied for the continuous tracking of detected objects. It is a robust algorithm that is convenient for real-time applications thanks to its fast run time speed of 20-30 frames per second (FPS) [16]. Overall, the objectives of this study were to utilize SSL for bringing more data into DL model development, add a new algorithm on top of the deep sort algorithm for solving the trajectory estimation and activity levels of individual broilers, and estimate flock level mobility.

## 2. Materials and Methods

The experiment was carried out in the Animal Science Department labs located in the Johnson Animal Research and Teaching Unit (JARTU), the University of Tennessee Knoxville, USA. It consisted of 28 chickens (Cobb 700, with a 1:1 male–female ratio) that were reared for a period of 54 days between October 18 and December 10, 2021. Day-old chicks were divided into 2 pens with 12 and 16 birds, respectively. We used birds at two growth rates: slow growing (<50 g/day) and standard (>65 g/day). Both groups of birds were reared under typical stocking densities, i.e., 24 kg/m^2^ for slow growing and 32 kg/m^2^ for standard, which translated to 12 birds/pen for slow-growing and 16 birds/pen for standard birds.

Each pen had a 100 cm × 150 cm pen with a standard camera (Amcrest UltraHD 5MP Outdoor POE Camera 2592 × 1944p) mounted at a 3 m height overlooking the pen, as depicted in Figure 1. The cameras recorded broiler movements continuously for 15 min per hour for 24 h. The 15 min period, i.e., the first quarter of every hour, was deemed statistically significant to represent the total mobility level of birds in an hour [17]. Additionally, it helped in effectively using computer storage while acquiring ample data for DL model development and mobility analysis. This setup collected data for 54 days continuously, encompassing the total life span of the broilers before they were moved to slaughterhouses at around 3 kg in weight. The video recordings were stored in standard hard drives for future analysis.

### 2.1. Semi-Supervised Learning

With time and human resource constraints, semi-supervised learning (SSL) is an influential method in tackling big datasets for DL model development. It lies between the supervised and unsupervised learning counterparts, benefiting from their respective strengths. SSL utilizes the guided learning procedure, i.e., learning from labeled datasets, the former method, and the unguided methodology, i.e., predicting and learning simultaneously from unlabeled datasets in the latter one [18,19].

In this study, firstly, to develop the initial YOLOv5 model, we employed a dataset with 1650 labeled images extracted from the recordings of 5 random days. The video recordings when the broilers were 7, 17, 26, 36, and 41 days of age have been used to build this dataset. Consequently, with a ratio of 1 frame per minute, images were extracted from them; this ratio was chosen so that we obtained images that were temporally further away from one another to reflect different possible positions of the broilers in the pens.

This dataset helped in providing a CNN model with a descent accuracy level, between 50 and 80%; later, it was applied to the next batch of 2160 unlabeled images extracted from the recordings and corresponded to broilers being 4, 18, 30, 44, and 47 days of age. Collectively, in these two steps, the dates were selected highly dispersed throughout the experimental period to utilize recordings that provided unique and less correlated information about broilers in their different rearing phases. They rendered a final trained model with high generalizability and lesser bias in recognizing broiler mobility. Eventually, the new successfully predicted images were fetched to the partially developed YOLOv5 model and made it learn further and increase its accuracy. Henceforth, we expected that it would help in creating a more robust and highly accurate model with minimal time and labor mobilization.

### 2.2. YOLOv5 DL Model

The novelty of the YOLOv5 model lies in its architecture. It comprises backbone, neck, and head sections. The backbone section focuses on distinguishing the receptive or feature fields in an image and tries to reduce the number of model parameters to achieve agility and lower memory requirement [20]. Meanwhile, the neck section classifies important image features, which improve the localization success rate. The head section combines low- and high-level features to increase precision rates and provides the final losses of the model [21].

We used three YOLOv5 model performance metrics, namely mean average precision (mAP), precision, and the regression coefficient of determination (R^2^), as depicted in Equations (1)–(3), respectively [22]. The mAP metric demonstrates the effectivity of the DL model by taking the mean of the average precision for each class under scrutiny. In this study, however, there was only one class, i.e., broiler, so the class number (Q) is equal to 1. Meanwhile, the precision metric is a measure of object detection and classification by the model, true positive is the successful detection of an object, and false positive shows misleading detections by the model. On the other hand, the R^2^ value shows how efficiently the model predicts (${\hat{y}}_{i}$) instances of actual data ($y_{i}$) [21].
(1)mAP=∑q=1QAP(q)Q
(2)$$Precision=\frac{True\;Positive}{True\;Positive+False\;Positive}$$
(3)$$R^{2}=1-\frac{\sum \left(y_{i}-{\hat{y}}_{i}\right)}{\sum {(y}_{i}-\hat{y})}$$

### 2.3. Deep Sort Algorithm

It is comprised of the Hungarian algorithm, the Kalman filter and the utilization of a bounding box, confidence, and deep features from frames detected by the CNN model. The Hungarian algorithm and the Kalman filter are applied for position and velocity parameter tracking and predict and update their future status accordingly. The integration of deep features extracted from the detected frames increases the likelihood of tracking objects effectively, even when they are located very near or are occluded [23,24]. As a result of consecutive frame association, motion prediction, and deep features, the tracking algorithm can attach identification (ID) numbers to each detected object and track them if it can differentiate them effectively. One problem arises during dense occlusion instances where this algorithm would not be able to associate objects in newer frames. Hence, it counts them as new objects and assigns new IDs. Here, we developed and applied a new algorithm integrated with the deep sort algorithm (neo-deep sort) that resolved or mitigated this problem.

### 2.4. Neo-Deep Sort Algorithm

The problem of occlusion or lost detection instances poses a major hindrance to calculating overall individual broiler mobilities. The YOLOv5 deep sort algorithm would resolve this issue for the most part. However, as the broiler mobility and stocking density increase, it becomes a monumental task to overcome this issue. Hence, we tested the following algorithm to correlate new and old IDs assigned on detected objects by the deep sort algorithm. Essentially, the algorithm would detect when a previously detected ID is lost and, consequently, it tries to identify when one/more new IDs appear in the deep sort algorithm results. It then processes several steps to correlate the new ID with the lost ID or delete those instances that are falsely detected; the algorithm flowchart is shown in Figure 2, and it was implemented using Python. The algorithm utilized the position of these lost and new objects to correlate them. If they were located at a distance smaller than the threshold distance, then the two IDs would be considered the same. This limit would be selected based on a trial-and-error process [25]. We validated this approach with manual ID correlation and/or deletion and the neo-deep sort results.

### 2.5. Final Model

The final model consists of the YOLOv5 model for object detection, the deep sort tracking algorithm, and the new algorithm to correlate and/or delete ID instances for solving the occlusion problem. We obtained the results after these 3 steps, and then the flock level and individual broiler’s displacements and their respective speeds were extracted and categorized accordingly.

The results of the final model consisted of individual broiler’s position in consecutive frames of a video. The model detected and tracked frames at an interval of about 1/100 s. Hence, even minuscule positional displacements of broilers were recorded. This can be an advantage of DL models and render challenging calculation burdens. Afterward, the coordinates of all the detected chickens in each frame were provided by the model in terms of maximum (x,y) values with the corresponding frame width and height. These values were used to assign a centroid coordinate (x_c_,y_c_) for each detected frame by Equation (4). The centroid values for consecutive frames were used to obtain a displacement value by the respective broiler. Eventually, the incremental displacement values would be added to obtain total displacements or moved distance of a broiler, as well as the flock mobility levels.

Broiler mobility determination was not a straightforward summation of displacements. We anticipated that small perturbations in body movements would result in the model reading the frames as a potential displacement. Therefore, we would try to validate the model’s results with respect to baseline manual measurements as well. This might lead us to some degree of calibration of the model that would eradicate the misleading results.
(4)xc,yc=x+xwidth2,y+yheight2

## 3. Results and Discussion

### 3.1. Broiler Pens and Data Collection

The broilers’ mobility was recorded for 15 min per hour daily for 54 days, rendering a total of 1268 recordings. These data were utilized for the training and development of the YOLOv5 model through the SSL approach. Eventually, broiler mobility analysis was carried out with the proposed YOLOv5 neo-deep sort model.

### 3.2. Semi-Supervised YOLOv5 Training

#### 3.2.1. Primary YOLOv5 Model Training

The initial training of the model with the manually labeled images resulted in the validation mAP, precision, and R^2^ levels, as shown in Figure 3. The graphs demonstrate a considerable increase in the success rates of the three metrics; though mAP, with an 81% success rate, may have more room for improvement. In other words, the precision and R^2^ values might be high since the model predicts only one class of object, i.e., chicken, while the mAP may require more training data to achieve a success rate above 90% with higher confidence levels. Overall, the primary trained YOLOv5 model could be considered effective enough to be employed on the unlabeled dataset to achieve one of our objectives.

#### 3.2.2. Labeling Images with the Primary YOLOv5 Model

The primary trained YOLOv5 model was used to predict labels for these images, which successfully predicted broilers in 2153 of them. A sample of detected images and the corresponding labels are depicted in Figure 4. It can be deduced that while SSL provides highly reliable results by pseudo-labeling images, it introduces some level of error to the model as well. But it is safe to say that the advantage of this method far outweighs its misleading results. Additionally, manual labeling of the first dataset took days to complete, while the pseudo-labeling of the images took less than an hour to complete. The SSL method was truly effective in providing ample labeled training data to further enhance the object detection success rate of the DL model.

#### 3.2.3. Final YOLOv5 Model Training

Finally, the model-labeled dataset was used to further train the existing trained YOLOv5 model. The newly labeled 2153 images were split into 80:20 sets for training and testing purposes, respectively. As a result of the second training, the YOLOv5 model’s predictive capabilities enhanced, as shown in Figure 5. It can be clearly seen that the mAP level has increased to 98% from the previous value of 81%. Henceforth, the trained YOLOv5 model has higher detection accuracy compared with the trained models in similar studies by Fang et al. [13] and Neethirajan [14]. Hence, it was now ready to be applied along with the neo-deep sort algorithm to analyze broiler hourly mobility levels.

### 3.3. Final Model: YOLOv5 Neo-Deep Sort Application

The final model was applied to the video recordings of the broilers from two separate cameras overlooking respective Pen #1 (twelve broilers) and Pen #2 (sixteen broilers). As discussed earlier, the data under study consist of recordings of when broilers were 11, 18, 24, 30, 36, 41, and 47 days old. Therefore, we were able to see the broiler mobility levels at different ages throughout the rearing process. It is worth mentioning that the model was trained by the data from Pen #2, and the resultant trained model was applied to one familiar environment, Pen #2, and a completely new environment, Pen #1. This would show the ability of the trained model to generalize and perform a completely new dataset.

#### 3.3.1. Broiler Detection Levels

The final model was able to detect broilers at different stages of their lifetime with reasonably high success rates. The general performance of our model is depicted in the distribution and boxplot graphs in Figure 6a–c. Overall, the number of broilers successfully detected in a frame shows a little skewed normal distribution with varying means of 9 and 14 for Pens 1 and 2, respectively. On average, the model was proportionate, 9/12 and 14/16, and was almost equally successful in detecting the number of broilers in both pens. But, in general, it performed better in Pen #2 over the course of 7 weeks. As seen in Figure 6c, the boxplots in the model were consistently performing better in detecting broilers in Pen #2 compared to Pen #1. Although they do not provide a higher success rate in the first batch of data from Pen #2, i.e., when the birds were 11 days old. They rendered better results in the consequent datapoints. On the other hand, in Pen #1, where the broiler stocking density is lower and the environment is newer to the model, the performance is comparatively less successful but still consistent and has reasonably high success rates.

#### 3.3.2. Broiler Tracking Performance

After the broiler detection process, the tracking step, i.e., the deep sort algorithm, tried to extract the movement coordinates of the broilers from consequent frames. In some instances, the tracking sequence of a particular broiler might be lost for a short time due to occlusions, broilers gliding over, or mixing closely with each other. After reappearing, the model would start tracking those birds again but would deem them as new birds and hence assign a new ID. Here, the final model’s continuous tracking ability of a broiler before losing it is shown in Figure 7a. On average, the model has tracked birds in different pens with varying success rates; it has tracked the birds better in Pen #2 compared to the results in Pen #1. In general, the model has tracked the broilers increasingly better as their ages grew. In other words, as broilers’ weight and volume increased, it was relatively easier for the model to continuously track them. On average, the broilers were tracked at least 3 min (20% of the 15 min) into the monitoring period. This may provide enough information on the speed, displacement, and mobility levels of a particular broiler, even if we do not try to associate different IDs of a particular broiler.

#### 3.3.3. Broiler Flock Mobility Level

The final model was utilized to track birds using the displacements between consecutive frames from video recordings separated by 1 s. This period was selected to decrease computation complexity by ignoring broiler perturbations that might have happened in less than a second. Consequently, the broilers’ total mobility, including moved distances and speed, was calculated. On the other hand, our proposed algorithm was used to associate different broiler IDs, as discussed in the previous sections. Henceforth, we were able to categorize broilers’ mobility comparatively efficiently at each hour of the day. This crucial result paved the way for continuous insights, enabling timely and effective interventions on birds with low or no mobility.

##### Total Displacement of Broilers at the Flock Level

The average daily covered distance by all the broilers and the corresponding average broiler weights in each pen are demonstrated in Figure 7b,c. As the broilers grew, their weights increased constantly until the saturation point at the end of the second month. But, the total displacement level had a different trend in both pens. The average daily displacements are highest when broilers are aged 18 d. Comparatively, Pen #2 has had a higher displacement level compared to Pen #1. It can be safely pointed out that higher stocking densities, such as in Pen #2, cause higher levels of mobility, as going from one point to another may require more walking and cause disturbance to the surrounding broilers. Additionally, the daily average covered distance trends give a better understanding of flock movements than in methods followed by similar works, such as Neethirajan’s [14], which have avoided providing a quantitative result for this level of movements.

##### Flock Speed Levels

The average daily broiler speed levels show a similar distribution as the displacements explained in the previous subsection. As indicated in Figure 8a., broilers in both pens have shown very similar speed levels, although the mean speed in Pen #2 (with 16 broilers) is higher than Pen #1 (with 12 broilers). The lowest average speed happened when broilers were 24 to 25 days of age in both pens. Meanwhile, the highest broiler speeds were recorded when the broilers were very young, around the age of 18 d. In both pens, the speed levels have increased steadily from the age of 25 d to the age of 45 d. On the other hand, the average speed changes compared to the increase in the weights of the birds demonstrate a similar trend. The average speed in both pens decreases to a minimum, with birds having an average weight of 1200 g–1400 g. As weight continues to increase, so does the speed of the birds as they reach 3000 g in weight. Pen #1 birds seemed to move slower while the other pen’s birds continued moving faster.

##### New Algorithm Application

The proposed algorithm was applied to associate different IDs that were assigned to the same broiler during the tracking. Figure 9a shows a sample hourly displacement tracking of the birds on a specific day; for example, at 6:00 a.m., as per the preliminary results, most of the broilers have moved moderately but birds 12, 14, and 16 have shown no or very low mobility. But, after the algorithm was applied, it was able to associate different broiler instances to give a final picture of the mobility of the birds. In this hour, in Figure 9a, IDs 6, 9, 10, 12, 14, and 16 were associated with IDs 126, 118, 108, 87, 115, and 146, respectively. The deep sort tracking algorithm had tracked the broilers with the former IDs and had lost their track due to occlusion instances. After resuming the tracking when the respective birds had reappeared, the new IDs, in the former list, were attributed to these broilers. Hence, with the proposed algorithm, we associated these lost instances. As a result, the final mobility level of the broilers is calculated by adding the individual displacements of the associated IDs. As seen in Figure 9b, the above-mentioned birds’ final displacement levels have changed after this process. For example, bird #6 had a displacement of about 10 cm, which increased to about 20 cm after ID association. As in the case of bird #14, even after ID association, the resultant mobility level and tracking appearance percentage still fell below 100%. Hence, we can conclude that even after the ID association process, we might not obtain a very high percentage of some broiler’s appearances. But, even a 50% appearance level can give us a statistically sound indication of how much a broiler is mobile. Additionally, we can further investigate the sequential videos of that day and obtain a bigger picture of that broiler. On the other hand, the final hourly speed and covered distance by individual broilers provide a more detailed insight into their mobility levels with higher accuracies compared with similar works, such as Neethirajan [14], Nasiri et al. [9], and Fang et al. [13]. This may help farm managers have more control over their resources and day-to-day interventions.

## 4. Conclusions

Broilers’ daily mobility levels significantly impact their daily behaviors, which could be detrimental to rendering high-quality meat to the market. In this study, we have utilized the semi-supervised YOLOv5 DL model alongside the deep sort algorithm for object detection and tracking, respectively. Additionally, we proposed a new algorithm that tackled deep sort tracking losses due to bird occlusion instances. A total of 7 days of hourly videos, corresponding to seven consecutive weeks, were studied. The SSL method paved the way to train the YOLOv5 model with considerably fewer human and time resources. It helped in increasing the YOLOv5 detection accuracy from 81% to 98%. The final YOLOv5 deep sort model was influential in tracking broilers continuously, at least around 20% of the time, but it had limitations in doing so during occlusion periods. Our proposed algorithm lessened this problem to some extent but was incapable of fully solving the tracking lost instances problem. Henceforth, in general, individual broiler and flock level displacements and speeds were computed throughout their growth period. This helped in understanding their mobility trends and categorizing them accordingly. This vision-based study of mobility indicators could be further developed by incorporating more effective algorithms to solve the lost information problem due to occlusions.

## Figures and Tables

**Figure 1 animals-13-02719-f001:**
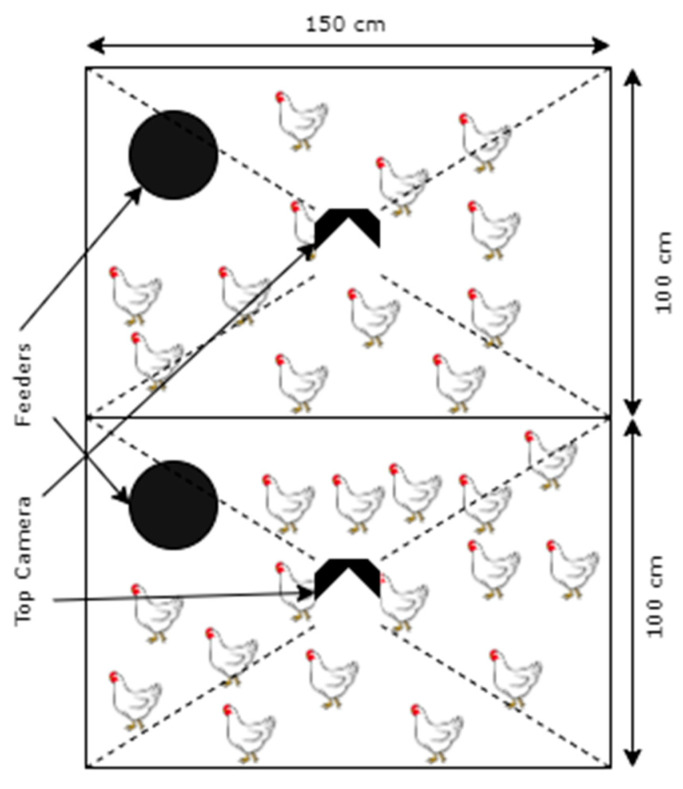
Pen dimensions, broilers, feeders, and camera positions.

**Figure 2 animals-13-02719-f002:**
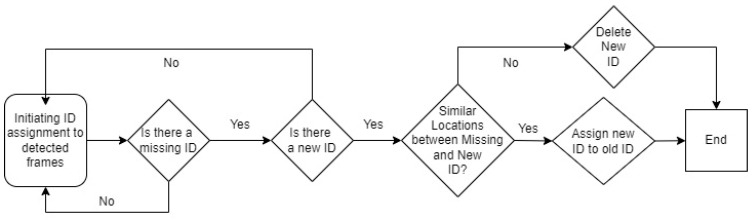
Neo-deep sort algorithm flowchart.

**Figure 3 animals-13-02719-f003:**
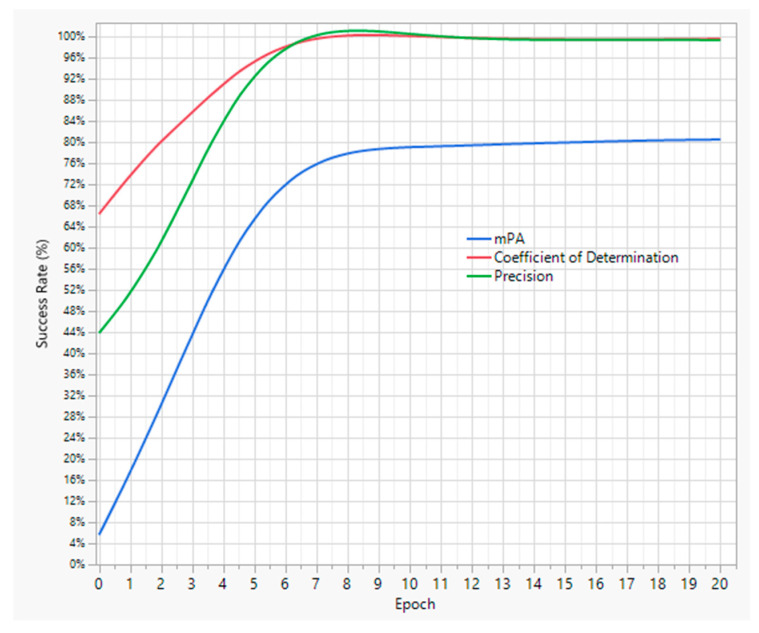
Training results of the primary YOLOv5 model.

**Figure 4 animals-13-02719-f004:**
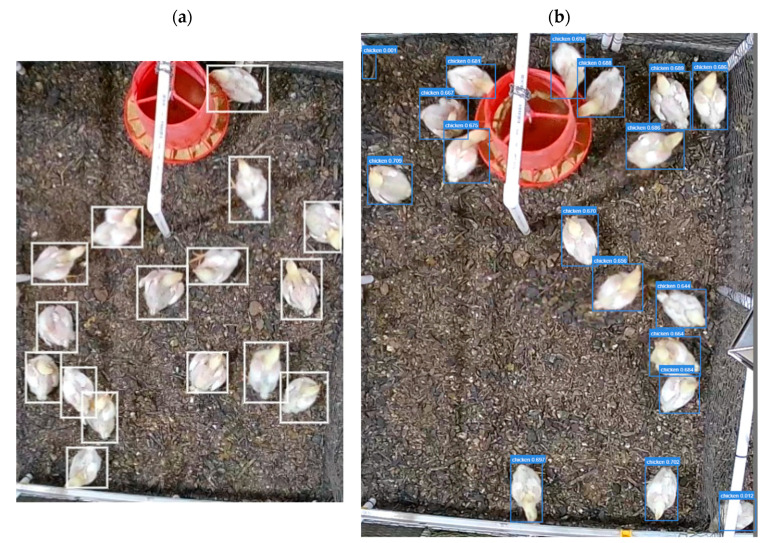
(**a**) A manually labeled image. (**b**) An image labeled by the YOLOv5 model.

**Figure 5 animals-13-02719-f005:**
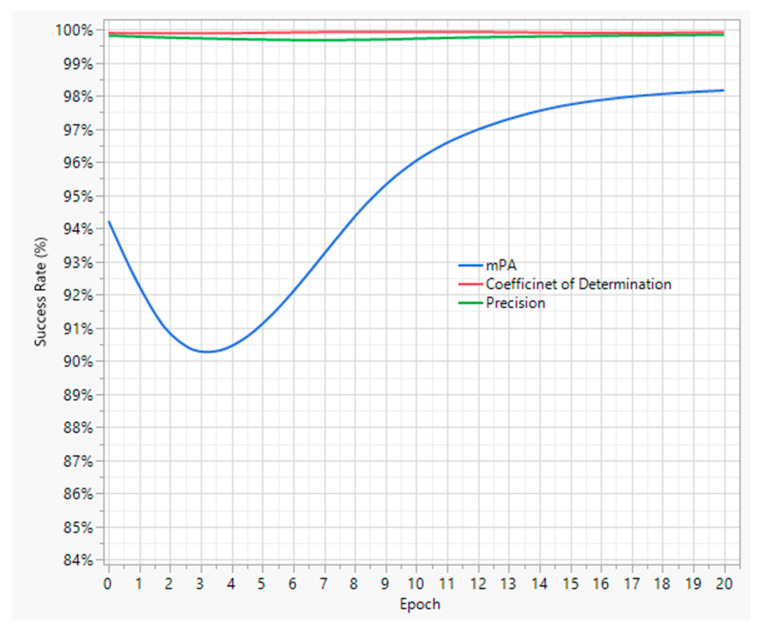
Second training results of the YOLOv5 model.

**Figure 6 animals-13-02719-f006:**
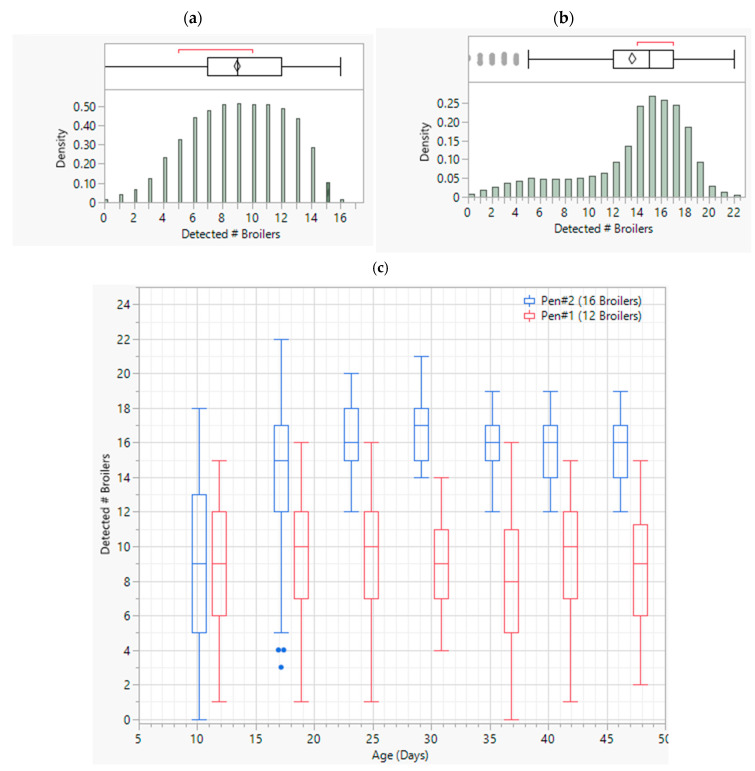
Broiler detection results by the final model. (**a**) Pen #1 frame success rate distribution. (**b**) Pen #2 frame success rate distribution. (**c**) Broiler success rate variation over different age periods.

**Figure 7 animals-13-02719-f007:**
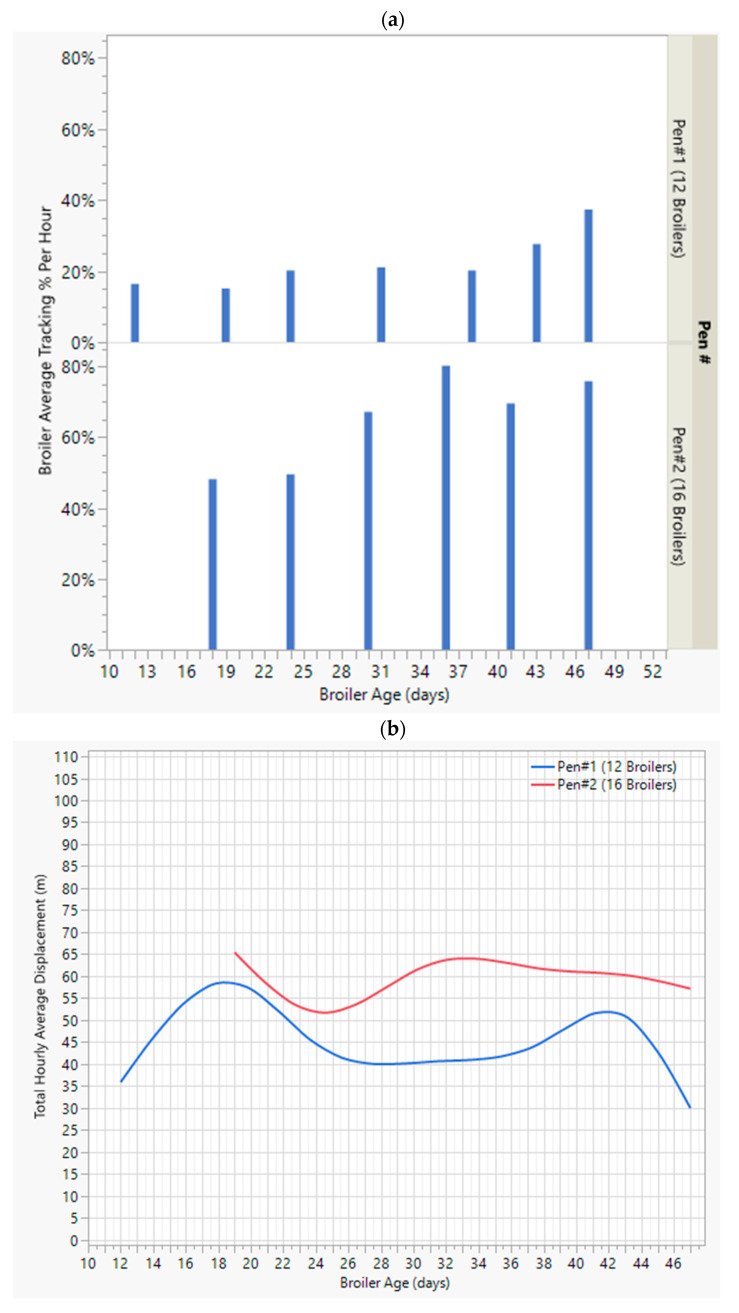
(**a**) Average broiler tracking level per hour. (**b**) Daily average displacement levels. (**c**) Average weight change in broilers as they grew.

**Figure 8 animals-13-02719-f008:**
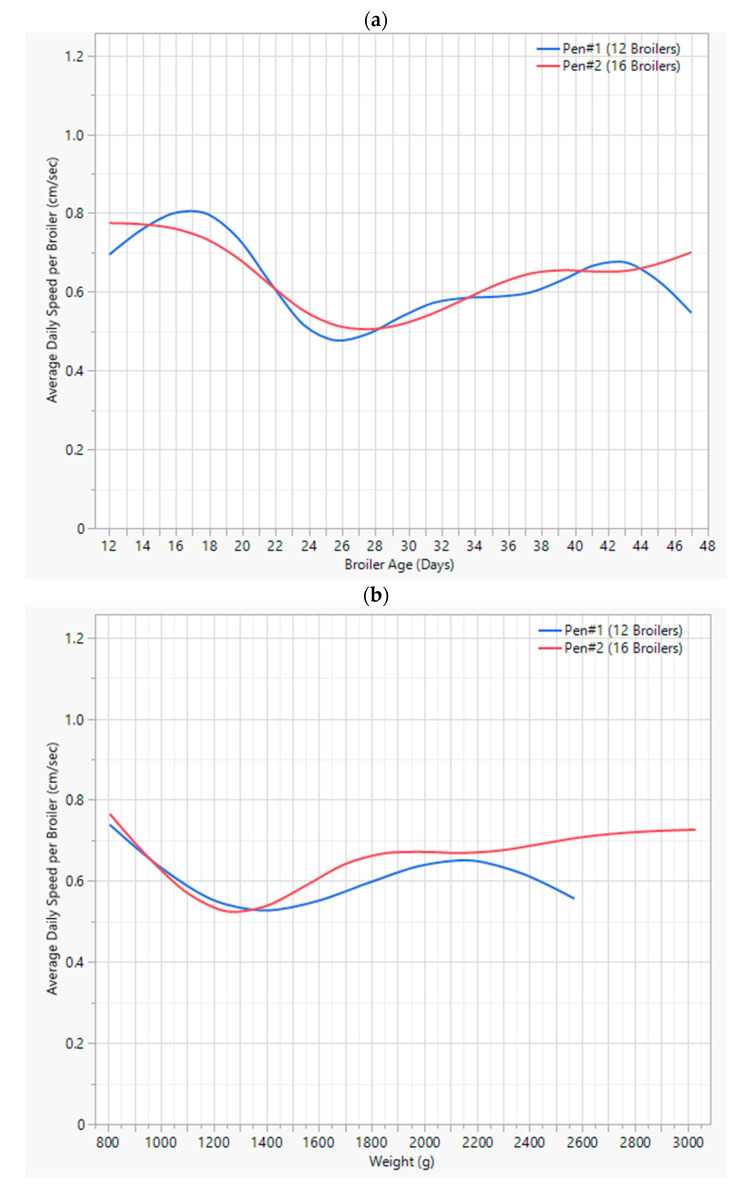
(**a**) Average speed change vs. broiler age. (**b**) Average speed change vs. weight.

**Figure 9 animals-13-02719-f009:**
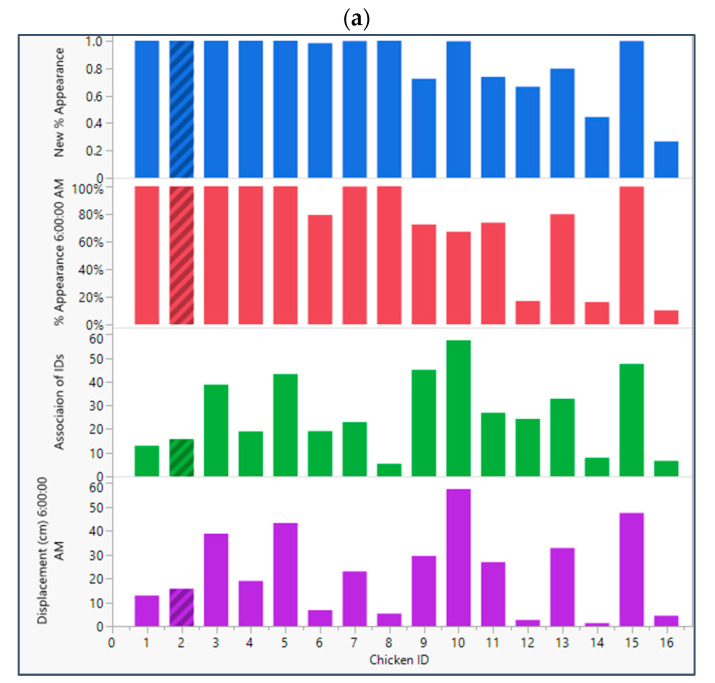
(**a**) Broilers’ ID association process and the respective changes in displacement and % appearance. (**b**) Individual broilers with their hourly mobility levels.

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
