# Peer review of "Broiler Mobility Assessment via a Semi-Supervised Deep Learning Model and Neo-Deep Sort Algorithm"

_animals, 2023, doi:10.3390/ani13172719_

Round 1

Reviewer 1 Report

Broiler Mobility Assessment Via a Semi-Supervised Deep 2 Learning Model and Neo-Deep Sort Algorithm

Manuscript ID: animals-2550926

Summary

The present study aimed to solve the estimation of the trajectory and activity levels of individual broilers, as well as to estimate mobility at flock level, by developing a new algorithm, using semi-supervised learning to bring more data into a deep learning model.

General concept comments

Article

It is an innovative study that provides important information to broiler welfare research, specifically focused on mobility or locomotion status.

Review

The manuscript is very well written, and the objective of the study can be clearly understood. Only a few changes need to be made to the paragraphs mentioned in the specific comments, and the wording of the references needs to be corrected.

Specific comments

61. “As in the works of [6-8] such features of individual birds are…”. It is understood that this applies to references 6 to 8. However, it is better if the authors are included in the paragraph.

69. “The authors in [9-11] have developed very…”. Same as point 61.

75. “[12] and [13] have applied a…”. Same as point 61.

78. “In the work of [14], a CNN based…”. Same as point 61.

100. On what basis was it decided to rear the chickens for 54 days? Why 54 days?

115-116. Although the dimensions of the pens can be assumed from the number of birds in figure 1, the dimensions mentioned in the title are not shown. Please consider this.

139-153. This entire paragraph should be placed in the Introduction.

140. You should present the meaning of the acronym YOLOv5 since the very first time it is mentioned in the manuscript.

167-178. This entire paragraph should be placed in the Introduction.

434-435. Please check the correct writing of this reference.

Reviewer 2 Report

In this paper, authors well described and propose a semi-supervised deep learning model and neo-deep sort algorithm to assess broiler mobility, combined these two model could provide individual boilers hourly moved distances accurately. Although this paper is well written and easily read, while several sections in the manuscript are less informative that require more details. Here are some comments and minor mistakes that the authors have to address.

1. Line: Part of “1. Introduction”, deep learning and artificial intelligence are the most studied topics, widely used in broiler motion recognition and tracking. It is strongly suggested to add more discussions about the specific research directions or topics using deep learning.

2. Line 108: it’s suggested to add details about 15 minute per hour. Is it the first 15 minutes or the last 15 minutes or others?

3. Line 126, Line 134: why do you choose these 5 days of age? There is still a difference between the two. Do they have the special effects for model?

4. Lines 387: “5. Conclusions” should be “4. Conclusions”.

5. Line 388: it’s unclear to get the conclusion of “Broiler’s daily mobility levels significantly impact their health conditions” through the whole part of “3. Results and Discussion”. Please revise the relevant description to emphasize this point.

Reviewer 3 Report

The work is very interesting and brings new elements regarding the assessment of the mobility of broiler chickens using artificial intelligence via a semi-supervised deep learning model and neo-deep sort algorithm.

On the basis of well-selected literature, the authors justified the purposefulness of the conducted research. The aim of the work was clearly formulated. The material used for the research is sufficient, the research methods have been selected appropriately and described in detail. The results are presented in 9 figures and discussed in detail.

Unfortunately, the authors did not compare their results with the results of similar studies dealing with the assessment of bird mobility. This should be supplemented.

The conclusions are correct and result from the obtained research results.
